# Radiation Graft-Copolymerization of Ultrafine Fully Vulcanized Powdered Natural Rubber: Effects of Styrene and Acrylonitrile Contents on Thermal Stability

**DOI:** 10.3390/polym13193447

**Published:** 2021-10-08

**Authors:** Niratchaporn Rimdusit, Chanchira Jubsilp, Phattarin Mora, Kasinee Hemvichian, Tran Thi Thuy, Panagiotis Karagiannidis, Sarawut Rimdusit

**Affiliations:** 1Research Unit in Polymeric Materials for Medical Practice Devices, Department of Chemical Engineering, Faculty of Engineering, Chulalongkorn University, Bangkok 10330, Thailand; R.njdearz@outlook.com (N.R.); Phattarin.M@gmail.com (P.M.); 2Department of Chemical Engineering, Faculty of Engineering, Srinakharinwirot University, Nakhonnayok 26120, Thailand; chanchira@g.swu.ac.th; 3Thailand Institute of Nuclear Technology, Bangkok 26120, Thailand; kasineeh@yahoo.com; 4Department of Analytical Chemistry, School of Chemical Engineering, Hanoi University of Science and Technology, Hanoi 100000, Vietnam; thuytranthi2708@gmail.com; 5School of Engineering, Faculty of Technology, University of Sunderland, Sunderland SR6 0DD, UK; Panagiotis.Karagiannidis@sunderland.ac.uk

**Keywords:** graft copolymer, natural rubber-graft-polystyrene, natural rubber-graft-polyacrylonitrile, ultrafine fully vulcanized powdered rubbers, electron beam vulcanization, spray drying

## Abstract

Graft copolymers, deproteinized natural rubber-graft-polystyrene (DPNR-g-PS) and deproteinized natural rubber-graft-polyacrylonitrile (DPNR-g-PAN), were prepared by the grafting of styrene (St) or acrylonitrile (AN) monomers onto DPNR latex via emulsion copolymerization. Then, ultrafine fully vulcanized powdered natural rubbers (UFPNRs) were produced by electron beam irradiation of the graft copolymers in the presence of di-trimethylolpropane tetra-acrylate (DTMPTA) as a crosslinking agent and, subsequently, a fast spray drying process. The effects of St or AN monomer contents and the radiation doses on the chemical structure, thermal stability, and physical properties of the graft copolymers and UFPNRs were investigated. The results showed that solvent resistance and grafting efficiency of DPNR-g-PS and DPNR-g-PAN were enhanced with increasing monomer content. SEM morphology of the UFPNRs showed separated and much less agglomerated particles with an average size about 6 μm. Therefore, it is possible that the developed UFPNRs grafted copolymers with good solvent resistance and rather high thermal stability can be used easily as toughening modifiers for polymers and their composites.

## 1. Introduction

Recently, ultrafine fully vulcanized powdered natural rubbers (UFPNRs) have been developed and demonstrated high potential as a toughening filler in a polymer matrix [1,2] with competitive performance compared to ultrafine fully vulcanized powdered rubbers (UFPRs) based on synthetic rubbers, such as styrene-butadiene (UFPSBR) [3,4], nitrile-butadiene (UFPNBR) [5,6] and polybutadiene rubber (UFPBR) [7].

UFPRs were prepared by irradiation of rubber latex (by gamma rays or electron beam) in the presence of crosslinking agent and subsequently a fast spray drying process. The obtained UFPRs have the same chemical composition as the feed [8,9]. An outstanding ability of the UFPRs is the fine dispersion and blending with other rubbers or other polymer matrices providing them good and balanced properties [10]. Compared with conventional rubber blends in a latex form, the UFPRs would provide some advantages for polymer blends, such as, for example, low energy consumed and time efficiency during the mixing process. This is because the interactions between crosslinked rubber particles are much lower than the cohesion of uncrosslinked rubber particles. Furthermore UFPRs provide a fine domain of the same size and good dispersion, no matter how high the blend ratio of UFPRs to matrix is, and depending on processing conditions and matrix types. Furthermore, all compositions could only be mixed into a continuous matrix phase because of the crosslinking of dispersed rubber that keeps an elastic state [10]. These characteristics have been reported in epoxy systems; a good dispersion between powdered rubber and epoxy resin leads to an improvement in the impact strength of epoxy resin being higher than that observed in an epoxy toughened with rubber latex [11]. In addition, UFPRs are used as toughening modifiers in various polymers, such as polybenzoxazine [6], nylon 6 [12,13], poly(lactic acid) [14,15], epoxy resin [11], PVC [16], and polypropylene [17,18,19] to improve their impact strength. In addition, the dynamic mechanical, mechanical and thermal properties were improved by adding a small quantity of UFPRs in polymer blends [11,12,13,14,17].

To prepare natural rubber (NR) in the UFPNRs form, it is first necessary to modify the properties of the NR before processing in order to reduce its high tackiness and enhance the compounding ability of the obtained UFPNRs. Moreover, NR lacks stability in non-polar solvents and organic solvents [20]. These drawbacks are typically solved by chemical modification and vulcanization. Grafting is one of the most effective chemical modification method; for example, grafting of a vinyl monomer with desirable properties. Styrene (St) [21], acrylonitrile (AN) [22], or methyl methacrylate (MMA) [23] was grafted onto a polyisoprene backbone to improve solvent resistance, thermal stability, mechanical properties, and compatibility of NR. Among the various used monomers, St and AN are among the most attractive monomers for grafting. St having a rigid aromatic benzene ring and similar polarity with NR showed grafting efficiency up to 70%. This behavior resulted in an improvement in thermal stability and mechanical properties of NR [20,24]. The high polarity of nitrile group of AN was found to enhance oil and organic solvents resistance along with mechanical properties and thermal stability [22,25,26,27,28]. In addition, radiation vulcanization, having some advantages, i.e., controlled crosslinking, fast process, clean technology, and fewer chemicals used [29] is a promising alternative against conventional methods. The radiation vulcanization uses high energy gamma rays or electron beam, without heating to promote the generation of free radicals in the linear main chain of NR and form a three-dimensional network of a thermoset. Consequently, softness and high tackiness of NR were addressed along with an increase in thermal stability and mechanical properties. Furthermore, the addition of polyfunctional monomers as a coagent in radiation vulcanization would help boost the vulcanization efficiency [30,31]. It has been reported that di-trimethylol propane tetraacrylate (DTMPTA) has high effectiveness in crosslinking of NR, due to its highly functionality (4 acrylate ester groups) and solubility in rubber latex [32].

Therefore, this work aims to prepare ultrafine fully vulcanized powdered natural rubbers (UFPNRs) by electron beam irradiation of graft copolymers in presence of DTMPTA coagent and subsequently a fast spray drying process. The graft copolymers were formulated by graft copolymerization of St or AN with deproteinized natural rubber (DPNR) via emulsion copolymerization, i.e., polystyrene grafted DPNR (DPNR-g-PS) and polyacrylonitrile grafted DPNR (DPNR-g-PAN). The effects of St or AN monomer contents, and radiation doses on chemical structure, physical and thermal properties of the graft copolymers and the UFPNRs were examined. The proper monomer content provided rather high thermal stability and an appropriate radiation dose to produce UFPNRs was obtained.

## 2. Materials and Methods

### 2.1. Experimental

#### Chemicals

High ammonia natural rubber latex (NR; 60% of dry rubber content (DRC)) was obtained from Sri Trang Agro-Industry Public Co., Ltd. (Bangkok, Thailand). Sodium dodecyl sulfate (SDS; 99%) from Merck Co., Ltd. (Darmstadt, Germany), urea (99.5%), magnesium-sulfate heptahydrate (MgSO_4_·7H_2_O), styrene monomer (99%), acrylonitrile monomer (99%), tert-butyl hydroperoxide (TBHPO; 70% in water), tetra-ethylene pentamine (TEPA; 94% in water), di-trimethylolpropane tetra-acrylate (DTMPTA), sodium hydroxide (NaOH), acetone (99.5%), and 2-butanone (99%) were purchased from Chemical Express Co., Ltd. (Bangkok, Thailand).

### 2.2. Preparation of Deproteinized Natural Rubber Latex (DPNR), St, and AN Monomers

DPNR was prepared by incubation of NR with 0.1 wt% urea and 1 wt% SDS at room temperature for 60 min, followed by centrifugation at 10,000 rpm and 15 °C for 30 min. The obtained cream fraction was redispersed in 0.5 wt% SDS solution and it was then centrifuged. After washing twice, the resulting DPNR contained about 0.4 wt% SDS to stabilize DPNR [17,20]. The protein content was measured by Kjeldahl analysis.

St and AN monomers were extracted with 10 wt%-aqueous sodium hydroxide solution and washed with de-ionized water until neutral to remove the inhibitor and then dried in MgSO_4_ [33].

### 2.3. Graft Copolymer Preparation

DPNR of 300 g, containing 30% DRC was added into a 500 cm^3^ glass reactor equipped with a mechanical stirrer. The DPNR was then bubbled with a nitrogen stream under stirring at 400 rpm and 30 °C for 2 h to remove the dissolved oxygen in the DPNR. Redox initiators, i.e., TEPA and TBHPO were dropped at a concentration of 3.5 × 10^−5^ mol/g of dry rubber and St or AN monomer was added to graft with DPNR at a content varied in a range of 5 to 20 phr. The grafting reaction was performed by stirring the DPNR at 400 rpm and 30 °C for 2 h to obtain DPNR-g-PS and DPNR-g-PAN at various St or AN monomer contents, respectively.

To determine monomer conversion and grafting efficiency, the ungrafted St or AN monomers were removed from the DPNR-g-PS and DPNR-g-PAN using a rotary evaporator at 80 °C for 45 min. Then, thin films of DNPR-g-PS and DPNR-g-PAN were prepared and dried in a vacuum oven at 50 °C for a week. The films were then purified by the Soxhlet extraction method to remove free homopolymers of PS or PAN and subsequently drying in vacuum oven at room temperature for a week.

After purification of DPNR-g-PS and DPNR-g-PAN, the monomer conversion and grafting efficiency were measured by gravimetric method, according to the expression below [20]:(1)$$\mathrm{Monomer}\;\mathrm{conversion}\;\left(\%\right)=\frac{\mathrm{Weight}\;\mathrm{of}\;\mathrm{polymer}\;\mathrm{in}\;\mathrm{gross}\;\mathrm{copolymer}}{\mathrm{Weight}\;\mathrm{of}\;\mathrm{monomer}}\times 100$$
(2)$$\mathrm{Grafting}\;\mathrm{efficiency}\;\left(\%\right)=\frac{\mathrm{Weight}\;\mathrm{of}\;\mathrm{polymer}\;\mathrm{linked}\;\mathrm{to}\;\mathrm{natural}\;\mathrm{rubber}}{\mathrm{Weight}\;\mathrm{of}\;\mathrm{polymer}\;\mathrm{in}\;\mathrm{gross}\;\mathrm{copolymer}}\times 100$$

### 2.4. Preparation of Ultrafine Fully Vulcanized Powdered Natural Rubber (UFPNRs)

The DPNR-g-PS and DPNR-g-PAN was diluted to 20% DRC with de-ionized water and mixed with 1 phr DTMPTA and then stirred for 15 min before crosslinked by electron beam irradiation at the doses of 50, 100, 200, and 300 kGy which are from electron energy of 10 MeV, beam power of 50 kW and a dose rate of 10 kGy/min. Then, the crosslinked DPNR-g-PS and DPNR-g-PAN was dried by a spray dryer (model B-290 from BUCHI, Switzerland) with the inlet temperature of 150 °C, feed flow rate of 7 mL/min, and air flow rate of 500 L/hr to achieve the UFPNRs in the final product.

### 2.5. Samples Characterization

The chemical structure of the DPNR-g-PS and DPNR-g-PAN films obtained after Soxhlet extraction and the UFPNRs were characterized using FTIR spectroscopy (model 2000 FTIR, Perkin Elmer) with an attenuated total reflection (ATR) accessory (Waltham, MA, United States) in the range of 400 to 4000 cm^−1^ by averaging 128 scans at a resolution of 4 cm^−1^. To qualitatively confirm the FTIR results, the structures of the graft copolymer swollen in CDCl_3_ were also analyzed using ^1^H-NMR spectroscopy recorded on a Bruker AV500D spectrometer 500 MHz (Bruker, Switzerland). All measurements were performed at 25 °C, using the pulse accumulation of 64 scans and LB parameter of 0.30 Hz.

Morphology and particle size of UFPNRs were investigated by scanning electron microscopy (SEM, model JSM-6510A from JEOL Ltd., Tokyo, Japan) with an accelerating voltage of 15 kV. The UFPNRs were coated with a thin layer of gold using a JEOL ion sputtering device (model JFC-1200) for 4 min.

The degradation temperature of NR film, DPNR-g-PS and DPNR-g-PAN films, and UFPNRs was evaluated using a thermogravimetric analyzer (model TGA1 Module Mettler-Toledo, Thailand). Samples of ~10 mg were heated from 25 to 850 °C with a heating rate of 10 °C/min under nitrogen atmosphere at a nitrogen purge gas of 50 mL/min.

The swelling properties and gel content of DPNR-g-PS and DPNR-g-PAN films, and UFPNRs were evaluated. The weight of sample (*W*_1_) was carefully measured before immersing in toluene (*ρ_s_* = 0.87 g/cm^3^, *V*_1_ = 106.5 mL/mol) for 24 h. After that, the swollen sample was immediately weighted (*W*_2_), followed by drying in a vacuum oven for 24 h at 70 °C to remove the solvent to obtain dried weight (*W*_3_). The swelling ratio (*Q*), molecular weight between crosslinks (*M_c_*), crosslink density (*CLD*) and gel fraction (*g*) were calculated by using the following Flory–Rehner equations [6] as can be seen in Equations (3)–(6), respectively.
*Q* = [(*W_2_* − *W_1_*)/*W_1_*] × [*ρ_r_*/*ρ_s_*](3)
*M_c_* = [−*ρ_r_V_1_*(*ϕ_r_*^1/3^ − *ϕ_r_*/2)]/[Ln(1 − *ϕ_r_*) + *ϕ_r_* + *χ*_12_*ϕ_r_*^2^](4)
where *ϕ_r_* = 1/(1 + *Q*)
*CLD* = *ρ_r_N*/*M_c_*
(5)
*g* = *W*_3_/*W*_1_
(6)
where *W*_1_, *W*_2_, and *W*_3_ are the weights of initial, swollen and dried samples, respectively; *ρ_s_* and *ρ_r_* is the density of solvent (0.87 g/cm^3^ of toluene) and rubber, respectively; *ϕ_r_* is the volume fraction of polymer in the swollen sample; *V*_1_ is the molar volume of the toluene solvent (106.5 mL/mol); *χ*_12_ is the polymer–solvent interaction parameter (the values of *χ*_12_ are 0.393 for toluene), and *N* is Avogadro’s number (6.022 × 10^23^).

The degree of crosslinking and chain scissions during the radiation process of UFPNRs were evaluated by sol-gel analysis of Charlesby Pinner equation [34];
*S* + *S*^0.5^ = (*p*_0_/*q*_0_) + (2/*q*_0_*u*_0_*D*)(7)
where *S* is a sol fraction (1—gel fraction); *p*_0_ is an average number of chain scissions per unit dose (kGy); *q*_0_ is an average number of crosslinking per unit dose (kGy); *u*_0_ is weight average degree of polymerization, and *D* is radiation dose (kGy).

## 3. Results and Discussion

### 3.1. Chemical Structure of Graft Copolymers

In the grafting process of DPNR, there are two possibilities for the grafting position of PS or PAN macroradicals on polyisoprene backbone, as shown in Figure 1. Firstly, a radical initiator (R·) reacts with a C=C bond of polyisoprene macromolecule and causes breaking of the C=C bond and simultaneously the formation of an active radical in the polyisoprene macromolecule. Then, a macroradical of PS or PAN can react with the radical formed in the polyisoprene macromolecule, as shown in Figure 1(a1,b1) (grafting). Secondly, a radical initiator (R·) can abstract a hydrogen atom from −CH_2_− and form an active site at the allylic carbon, which is the carbon next to the double bond of polyisoprene macromolecules. Thus, PS or PAN macroradicals can attach to the active site of allylic carbon to form graft copolymer [35], as shown in Figure 1(a2,b2), respectively.

To confirm the grafted position of PS or PAN on polyisoprene backbone, the FTIR spectra of DPNR, DPNR-g-PS, and DPNR-g-PAN were obtained and are presented in Figure 2. All samples showed the characteristic absorbance peaks of *cis*-1,4-polyisoprene, at 2961 and 2852 cm^−1^ corresponding to C−H stretching of CH_3_ and CH_2_, respectively; also at 1448 and 1376 cm^−1^ attributed to C−H deformation of −CH_2_ and of −CH−, respectively [36,37,38]. In addition, C=C stretching vibration at slope of 1600 cm^−1^ and the peak of 836 cm^−1^ due to C=C bending vibration of DPNR main chain were observed. Regarding DPNR-g-PS (Figure 2b), the new absorbance peaks of 1542 and 760 cm^−1^ related to C=C stretching vibration of the phenyl ring, and 698 cm^−1^ attributed to mono substituted benzyl ring of styrene structure were observed [25]. Although, in Figure 2c, NR-g-PAN showed the new absorbance peak at 2253 cm^−1^ related to nitrile (C≡N) stretching of acrylonitrile [24]. Furthermore, the intensity of absorbance peak can imply the quantity of functional groups present in graft copolymer. The lower intensity peaks at 1600 and 836 cm^−1^ of DPNR-g-PS and DPNR-g-PAN compared with DPNR were attributed to consumption of double bonds during grafting process.

To qualitatively confirm the FTIR results, the ^1^H NMR of graft copolymers is presented in Figure 3. The ^1^H NMR spectrum of DPNR-g-PS (Figure 3a) exhibited the unsaturated methyne proton of DPNR at a resonance of 5.1 ppm (olefinic proton) (a), the peak at 2.0 ppm attributed to methylene protons (b, b′) [25,39]. The peak at 1.7 ppm approved for methyl proton (c) of DPNR was also observed. Moreover, the peaks at 7.3 ppm attributed to the aromatic protons of St (d, d′) were found [40] and the peak at 1.3–1.4 ppm indicating the methylene protons (e) of DPNR linked to St in DPNR-g-PS was obtained [40]. For the ^1^H NMR spectrum of DPNR-g-PAN (Figure 3b), the peak of DPNR at 5.1 ppm (olefinic proton), at 2.0 ppm (−CH_2_) and 1.7 ppm (−CH_3_) were also observed. The peak at 1.3–1.4 ppm attributed to methylene protons of DPNR linked to AN in DPNR-g-PAN was also detected. These results confirmed that the PS and PAN macroradicals were grafted onto the DPNR structure.

### 3.2. Effect of St and AN Monomer Content on Monomer Conversion and Grafting Efficiency of Graft Copolymers

Monomer conversion and grafting efficiency are the factors which evaluate the efficiency of the grafting process. The monomer conversion is defined as the percentage of monomer molecules converted to copolymer on polyisoprene chain including the free homopolymer presented in the aqueous phase. The grafting efficiency refers to percentage of monomer grafted onto the backbone of polyisoprene. The effect of St and AN content on monomer conversion and grafting efficiency of DPNR-g-PS and DPNR-g-PAN were evaluated by the gravimetric method. Figure 4a shows that the monomer conversion of St significantly increased from 47% up to 71% and then decreased to 55% at St content 5, 15, and 20 phr, respectively. The grafting efficiency of PS onto DPNR was increased from 18% to 49%, 69% and then decreased to 60% for the St contents of 5, 10, 15, or 20 phr, respectively. Figure 4b shows that the addition of AN monomer at 5, 10, or 15 phr resulted in an increase in monomer conversion of 15%, 22%, or 46%, and then decreased to 29% at 20 phr of AN, while the grafting efficiency was increased with increasing AN contents up to 15 phr, i.e., 12%, 22%, 33%, or 24% for 5, 10, 15, or 20 phr, respectively. The addition of St or AN monomers more than 15 phr caused a reduction in both monomer conversion and grafting efficiency. These results may be explained by a surface-controlled process; that is the copolymerization occurred only inside the micelle on the rubber particle surface [41], resulting in formation of graft copolymer onto rubber surface. Additionally, homopolymer PS and PAN are formed in aqueous phase [35]. Thus, an increase in monomer content raised the chance of the reaction between monomer macroradicals with polyisoprenyl radicals and the formation of graft copolymer. However, when the shell thickness of graft copolymer reached the second stage, it led to a decrease in contact area between monomer and rubber. It is more difficult for monomers to diffuse the rubber chain and there were not enough active sites on the rubber surface for the newly arriving monomer macroradicals. This behavior resulted in more predominant PS or PAN formation than graft copolymers beyond 15 phr of monomer content. In addition, the grafting of DPNR with PAN provided monomer conversion and grafting efficiency lower than grafting with PS. This is due to high polarity of AN monomer leading to lower absorption into micelle, including non-polar DPNR and hydrophilic radical initiator, to react with DPNR, while the non-polar St monomers were higher absorbed.

### 3.3. Effect of St and AN Monomer Content on Thermal Stability of Graft Copolymers

Thermal stability of NR and graft copolymers were characterized by the degradation temperature at 5% weight loss (T_d5_). From the previous section, the grafting efficiency and monomer conversion were decreased with the attempt to add St and AN monomers more than 15 phr. In this section, the effect of St and AN contents in the range of 5 to 15 phr on thermal stability of graft copolymer was continuously investigated. The effect of St content on the T_d5_ of DPNR-g-PS is plotted in Figure 5a. Grafting improved T_d5_ from 334 °C for NR to 342, 345, and 347 °C for PS content of 5, 10, and 15 phr, respectively, an improvement of 8–13 °C compared to that of NR was obtained. This is due to the interaction of PS and DPNR, along with the high stability of free PS containing aromatic benzene ring. The latter has high degradation temperature of 375–450 °C [42]. However, the T_d5_ of DPNR-g-PS with of St content of 20 phr is slightly less than 347 °C, due to the lower grafting efficiency as previously mentioned.

In DPNR-g-PAN, the effects of AN contents on T_d5_ are shown in Figure 5b. The grafting of 5 phr PAN onto DPNR showed T_d5_ of 337 °C. This is due to the dipole–dipole interactions of polar functional groups of PAN chains grafted onto DPNR backbone, which are stronger than those of London interaction of non-polar functional groups of DPNR. This behavior led to higher decomposition temperature of DPNR-g-PAN. However, the T_d5_s of DPNR-g-PAN at various AN contents slightly increased, i.e., 337, 338, 339, and 340 °C for DPNR grafted with 5, 10, 15, and 20 phr of AN, respectively. This result indicates that the low grafting efficiency of PAN resulted in insignificant difference of individual grafted PAN content. In addition, non-rigidity of PAN structure did not cause any significant improvement of DPNR’s thermal stability [43].

### 3.4. Effect of St and AN Monomer Content on Swelling Behavior of Graft Copolymers

Swelling behavior of a polymer in a certain solvent is a parameter indicating the ability of solvent to penetrate into void space of the polymeric chain network. According to the Hansen solubility parameter theory, high polymer-solvent compatibility due to similar polarity and solubility parameter between polymer and solvent causes the solvent diffusivity into the polymer network. This behavior leads to swelling and change in physical and chemical properties of the polymer that result in a decreased efficiency during polymer applications [27,44]. The parameters which express the swelling behavior are swelling ratio (Q), molecular weight between crosslinks (M_c_), crosslink density (CLD) and gel fraction (g). The effect of St monomer content on swelling behavior of DPNR-g-PS in toluene is plotted in Figure 6. Figure 6a showed that the swelling ratio of all DPNR-g-PSs is lower than that of DPNR. Additionally, the swelling ratio of all NR-g-PSs decreased with increasing the St content up to 15 phr, while, the gel fraction of all NR-g-PSs is higher than that of DPNR and slightly increased with the increase in the St monomer until 15 phr. It is possible that the presence of PS increases the solubility parameter of DPNR, which leads to power of solvent resistance [27]. Moreover, during grafting process, the presence of TEPA/TBHPO as initiator and cross-linker generated active sites on DPNR chain and then to form a physical crosslinked network which restrict the diffusion of solvent that resulted in lower swelling ratio and higher gel content [45]. However, the addition of St monomer above 15 phr caused an increase in the swelling ratio of DPNR-g-PS and a decrease in the gel fraction due to lower grafting efficiency, which led to reduced solvent resistance. In Figure 6b, an increase in the crosslink density of the DPNR-g-PSs with increasing St monomer up to 15 phr was observed. This increase in crosslink density hindered solvent penetration into graft copolymers and led to a decrease in swelling ratio. Moreover, as expected, the molecular weight between the crosslinks of DPNR-g-PSs decreased up to St monomer of 15 phr, and then increased for St monomer of 20 phr. All swelling results showed that grafting with PS improved the solvent resistance of DPNR.

The swelling behavior of DPNR-g-PANs showed the same trend with DPNR-g-PSs. That is, a decrease in swelling ratio and an increase in gel fraction of the DPNR-g-PAN with increasing AN monomer up to 15 phr, as shown in Figure 7a. This suggests that the interactions between the high polarity functional groups of PAN and DPNR chain decrease the van der Waals interactions with toluene. Thus, a physical crosslinked network is formed during grafting which restricts the diffusion of toluene into the graft copolymer. These characteristics rendered lower swelling ratio and higher gel content. However, the addition of AN monomer above 15 phr resulted in an increase the swelling ratio and a decrease in the gel fraction. The crosslink density (Figure 7b), was found to increase with increasing AN monomers up to 15 phr, while the DPNR-g-PAN (15 phr) showed the lowest molecular weight between crosslinks. This behavior resulted in hindrance of solvent penetration, and thus a decrease in swelling ratio was obtained. Therefore, grafting with PAN also improved the solvent resistance of DPNR.

### 3.5. Morphology of Powdered Rubber of Graft Copolymers

The morphology of powdered rubbers of NR, DPNR-g-PS, and DPNR-g-PAN was investigated by SEM. In Figure 8a, agglomeration and high tackiness particles of the NR were observed. The DPNR-g-PS (15 phr) and DPNR-g-PAN (15 phr) particles, which had the highest grafting efficiency, still exhibit agglomeration after spray drying (Figure 8b,c). It is possible that grafting with PS or PAN did not cause sufficient crosslinks between DPNR macromolecules to reduce tackiness of rubber particles and thus being able to measure the particle size of grafted powdered rubber. This suggests that only suitable high crosslink density of rubber lattice could produce practical UFPNRs [3]. Therefore, the DPNR-g-PS and DPNR-g-PAN need to be vulcanized to promote the increase in crosslink density of three-dimensional network before producing UFPNRs.

### 3.6. Chemical Structure of UFPNR-g-PS and UFPNR-g-PAN

UFPNR-g-PS and UFPNR-g-PAN were prepared by irradiation of DPNR-g-PS and DPNR-g-PAN by electron beam in the presence of DTMPTA to form a three-dimensional crosslinked network as proposed in Figure 9. The probable structure of crosslinked UFPNR-g-PS and UFPNR-g-PAN is shown in Figure 9b,c. The chemical structure of UFPNRs obtained with various irradiation doses were investigated by FTIR spectroscopy. The FTIR spectra of the UFPNR-g-PS (5 phr) still showed the appearance of absorbance peaks at 1675–1640 cm^−1^ and 836 cm^−1^ attributed to C=C stretching vibration and C=C bending of DPNR main chain as seen in Figure 10. However, the intensity of peak at 836 cm^−1^ of the UFPNR-g-PS was higher at the higher radiation doses, indicating that the consumption of double bonds decreased because of the abundant energy radiation leading to more chain scission reaction [34]. The absorbance peak at 1080 cm^−1^ attributed to formation of C−C in crosslinked network was also lower at higher radiation doses. The absorbance peak in range of 1755 to 1745 cm^−1^ is due to the of ester group of DTMPTA.

Figure 11 shows the FTIR spectra of UFPNR-g-PAN (5 phr) with various radiation doses. The signals of C=C stretching vibration and C=C bending of DPNR main chain, C−C in crosslink network between coagent and DPNR main chain were observed at 1675–1640 cm^−1^, 836 cm^−1^, and 1080 cm^−1^, respectively. Moreover, the intensity of absorbance peak at 836 cm^−1^ of the UFPNR-g-PAN at lower irradiation dose is lower than that of the UFPNR-g-PAN at higher radiation doses, indicating breaking of double bonds to form crosslinking. In addition, after irradiation for all radiation doses, the absorbance peak of C≡N group at 2253 cm^−1^ disappeared, due to its conversion to C=N with a stretching peak in a range of 1690 to 1640 cm^−1^ [46]. The presence of ester group of DTMPTA was also observed at 1755–1745 cm^−1^.

### 3.7. Effect of St and AN Monomer Content and Radiation Dose on Thermal Stability of UFPNR-g-PS and UFPNR-g-PAN

The effect of monomer content on degradation temperature at 5% weight loss (T_d5_) of UFPNRs under 100 kGy was investigated. Figure 12a showed T_d5_, i.e., 346 °C, 344 °C, and 345 °C of UFPNR-g-PS at St monomer of 5, 10, and 15 phr, respectively. It was observed that T_d5_ of the UFPNR-g-PS tended to decrease with an increase in St monomer content. This implies that in the radiation vulcanization process, two types of reactions, i.e., crosslinking and chain scission, occur simultaneously and randomly, causing changes in polymer structure and its properties. For crosslinking, the radicals are generated at the double bond of rubber main chain by radiation and recombined to neighbor radical chains. This leads to the formation of C−C crosslinks of three-dimension network between rubber chains, leading to enhancement of rubber properties. Meanwhile, if a macroradical could not find neighbor macroradicals to recombine, the chain scission reaction is taking place that leads to decrease in thermal stability. Due to the high grafting efficiency, i.e., 70% for the DPNR-g-PS (15 phr), a great number of double bonds on the rubber chain were consumed to graft with PS chain; hence, the active site at double bond of rubber main chain was difficult to find adjacent macroradicals to crosslink, resulting in fragment of the long chains. Meanwhile, the DPNR-g-PS obtained by grafting at lower St monomer content results in less grafting efficiency and lower double bond consumption on the NR chains. Consequently, residual double bonds on NR chains are sufficient to promote more crosslinking with neighbor macroradicals. The aromatic benzene ring with a high stabilized resonance structure of St monomer also provides higher radiation resistance to rubber chain [47]. When exposed to high energy radiation, the St aromatic rings directly absorbed the energy and dissipate it; moreover, the radical on the rubber chain can be transferred to the rings and then decomposed without changing of the DPNR-g-PS structure. However, the benzene rings with high stiffness of PS and low flexibility of rubber chain resulted in block a partner macroradical to recombine, in consequent, chain scission of rubber chain was obtained which resulted in a decrease in thermal stability of DPNR-g-PS with increasing St monomer content [34,48]. As the reason mentioned above, the DPNR-g-PS (5 phr) exhibited rather high thermal stability, i.e., 346 °C which is significantly higher than that of the NR, i.e., 337 °C under 100 kGy [6]. Consequently, thermal stability of UFPNR-g-PS (5 phr) was studied at various radiation doses as depicted in Figure 12b. The T_d5_ of UFPNR-g-PS (5 phr) at 0, 50, 100, 200, and 300 kGy was 342, 346, 346, 344, and 343 °C, respectively. The trend of T_d5_ increase in the whole range is due to formation of a three-dimensional crosslinked network under the radiation influence. The higher T_d5_ of the UFPNR-g-PS (5 phr) at lower radiation dose, i.e., 50 and 100 kGy was observed. This behavior is probably due to the fact that the radiation dose of 50 kGy did not provide enough energy to promote more crosslinking, while the chain scission occurred to some extent with higher dose of 100 kGy. The thermal stability reached the optimum value at 50–100 kGy, because the absorbed energy promoted more crosslinking than chain scission. The increase in radiation dose beyond 100 kGy resulted in a decrease in T_d5_ of the UFPNR-g-PS (5 phr), because the abundant energy radiation leads to chain scission. As a result, shorter fragment chains were formed which could react with others or depolymerized [34].

The effect of AN monomer content on T_d5_ of UFPNR-g-PAN irradiated with 100 kGy was also investigated as depicted in Figure 13a. The T_d5_ of the UFPNR-g-PAN at AN monomer of 5, 10, and 15 phr was 339, 338, and 338 °C, respectively. This indicated that the AN monomer in a range of 5 to 15 phr showed no significantly affect on the T_d5_ of UFPNR-g-PAN because of the narrow grafting efficiency and the low grafting efficiency of DPNR-g-PAN. DPNR-g-PAN (5 phr) with T_d5_ of 339 °C was selected to evaluate its T_d5_ at the radiation dose in a range of 50 to 300 kGy (Figure 13b). It was found that the T_d5_ of un-irradiated UFPNR-g-PAN (5 phr) was 337 °C, while the UFPNR-g-PAN irradiated at 50–300 kGy showed more than 2–4 °C above the T_d5_ of the un-irradiated UFPNR-g-PAN, i.e., 339 °C at 100 kGy, 340 °C at 200 kGy, and 341 °C for 300 kGy. This indicated that the additional high energy radiation doses have sufficient energy to form more radicals on the rubber chain, thus more crosslinking reactions and formation of a three-dimensional crosslinking network was obtained. As previous mentioned in FTIR study, it was found that nitrile groups C≡N of PAN were broken to from C=N and crosslinked with others. This formation of a crosslinked network may lead to enhancement of T_d5_ of UFPNR-g-PAN [6].

### 3.8. Effect of Radiation Dose on Swelling Behavior of UFPNR-g-PS and UFPNR-g-PAN

One of the most important experimental parameters to produce UFPNR in this work is the radiation dose by electron beam. The effect of radiation dose on swelling behavior of UFPNR-g-PS (5 phr) is plotted in Figure 14a,b. In Figure 14a, the results show a significant reduction in swelling ratio of the UFPNR-g-PS at the whole radiation doses compared with the un-irradiated UFPNR-g-PS, while the gel fraction increased with increasing radiation doses. These characteristics may be due to radiation-induced crosslinking of UFPNR-g-PS. Although the chain scission occurred during radiation of the UFPNR-g-PS (5 phr) as mentioned in Section 3.7, the chain fragments can further react with other macroradicals to form crosslinks of short chains. Hence, the crosslinked network of the UFPNR-g-PS (5 phr) at various radiation doses caused strong chemical bonds between rubber chains, resulting in lower swelling ratio. This behavior is consistent with increment of gel fraction, indicating the enhancement of solvent resistance. In addition, as expected, the obtained UFPNR-g-PS (5 phr) at higher energy radiation doses showed more crosslinking that related with the decreasing of the molecular weight between the crosslinks as presented in Figure 14b.

The reduction in swelling ratio along with the increment of gel fraction, and the increase in crosslink density along with the decrease in the molecular weight between crosslinks at various radiation doses up to 300 kGy was also observed for UFPNR-g-PANs (5 phr) as plotted in Figure 15a,b, respectively. Moreover, at the same radiation dose, it can be observed that the swelling behavior of UFPNR-g-PS was lower than that of the UFPNR-g-PAN due to much lower grafting efficiency of AN monomer onto rubber structure, resulting in lower solvent resistance of the UFPNR-g-PAN.

The crosslinking and chain scission reactions occur simultaneously during the radiation process and influence the properties of UFPNRs. The sol-gel analysis of radiation crosslinking evaluated by using Equation (7) [34] for the UFPNR-g-PS (5 phr) and UFPNR-g-PAN (5 phr) at 50, 100, 200, and 300 kGy is plotted in Figure 16. The ratio of chain scission to crosslinking (*p*_0_*/q*_0_) obtained from the intercept of linear curve fitting of the data obtained for the UFPNR-g-PSs (5 phr) and the UFPNR-g-PANs (5 phr) were 0.5377 and 0.3942, respectively. These values show that the crosslinking reaction is accompanied randomly by rubber chain scission; the crosslinking tends to dominate chain scission in the whole radiation doses [6,34]. The number of crosslinking of the UFPNR-g-PS (5 phr) was lower than that of UFPNR-g-PAN (5 phr) in this radiation dose range due to radiation resistance of aromatic benzene ring in PS. However, deviation from a linear behavior is observed at the higher doses. This is due to more occurrence of crosslinking at higher radiation doses. Consequently, the more crosslinked chains restrict the motion of macromolecules. In addition, rubber chains were also degraded into fragment chains at higher radiation doses [6].

### 3.9. Effect of Radiation Dose on Morphology and Particle Sizes of UFPNR-g-PS and UFPNR-g-PAN

Figure 17 shows SEM micrographs of the UFPNR-g-PS (5 phr) irradiated at 50, 100, 200, and 300 kGy. The SEM micrographs show more separate particles at radiation dose of 50 kGy (Figure 17a) compared with un-irradiated DPNR-g-PS (5 phr) particles (Figure 8b). However, the images still show some agglomerated and imperfect spherical particles with rough surface. When the radiation dose increased from 100 kGy to 300 kGy (Figure 17b–d), very smooth spherical surfaces were observed. The agglomerated particles and the particle size tend to decrease at higher radiation doses. Moreover, the sample irradiated with 300 kGy showed the least aggregation along with better particle dispersion. The average particle size of the UFPNR-g-PSs was measured from 300 particles, and found to reduce from 10.64 ± 3.98 µm to 7.33 ± 3.49 µm, 6.15 ± 3.48 µm, and 5.95 ± 3.03 µm with increasing radiation doses from 50 kGy to 100, 200, and 300 kGy, respectively. It is possible that an influence of radiation doses on formation of a more crosslinked network and on modification the rubber surface could reduce the tackiness of rubber lattices leading to formation of lower agglomerated and smoother surface particles along with smaller particle sizes [7].

Figure 18 shows SEM micrographs of the UFPNR-g-PAN (5 phr) irradiated at 50, 100, 200 and 300 kGy. The morphology of UFPNR-g-PANs (5 phr) in the whole range of radiation doses shows more separated particles compared with the un-irradiated DPNR-g-PAN (5 phr) particles (Figure 8c). However, agglomerated, high tackiness and imperfect spherical particles at dose of 50 kGy are observed (Figure 18a). The smoother spherical surfaces along with less aggregation and more dispersed particles of UFPNR-g-PAN (5 phr), were observed with increasing the radiation dose in a range of 100 to 300 kGy (Figure 18b–d). This is due to formation of more crosslinked network after radiation which reduce the tackiness of rubber lattices [7]. Furthermore, it was found that the average particle size of the UFPNR-g-PANs was from 7.73 ± 3.29 µm to 6.95 ± 3.33 µm, 6.62 ± 2.90 µm and 6.39 ± 2.71 µm with increasing radiation doses from 50 kGy to 300 kGy, respectively. Regardless, the developed UFPNRs grafted St or AN monomers and subsequently irradiated at 300 kGy showed good dispersion the least aggregation than the UFPNR without grafting and irradiated approximately 350 kGy, however, the similar rubber particle size about 6 µm was obtained [1,2].

## 4. Conclusions

The graft copolymers, i.e., DPNR-g-PS and DPNR-g-PAN, were successfully prepared as confirmed by the study of their chemical structures using FTIR spectroscopy and ^1^H NMR. The increasing monomer content up to 15 phr provided the highest grafting efficiency of graft copolymers, in consequence, thermal stability, swelling ratio, and crosslink density of the graft copolymers were significantly enhanced. However, they could not be used for the production of powdered natural rubber after spray drying (aiming to reduction in tackiness of grafted DPNR lattices) due to their insufficient crosslink density. The DPNR-g-PS and the DPNR-g-PAN were then irradiated by electron beam to produce the practical UFPNR-g-PS and UFPNR-g-PAN through spray drying. It was found that the residual double bonds in DPNR at 5 phr St or 5 phr AN monomer contents were sufficient to generate more radicals to promote the formation of a crosslinking network under the same radiation dose of both UFPNRs, resulting in an improve of thermal stability. Moreover, the increment of radiation doses up to 300 kGy improved the thermal stability of both UFPNRs, i.e., 343 °C for UFPNR-g-PS (5 phr) and 341 °C for UFPNR-g-PAN (5 phr), compared with 334 °C of the unmodified NR. Furthermore, morphology study by SEM of the obtained UFPNRs grafted with 5 phr monomer content showed the good separated and much less agglomerated particles. The UFPNRs irradiated by radiation dose of 300 kGy showed particles with the smoothest spherical surface, the least amount of agglomerated particles with an average particle size about 6 μm. The results suggest that the proper monomer content of 5 phr and proper radiation dose of 300 kGy for producing UFPNR-g-PS and UFPNR-g-PAN with maintaining rather high thermal stability have a potential to be toughening filler in thermoplastic and thermoset polymers, and their polymer composites.

## Figures and Tables

**Figure 1 polymers-13-03447-f001:**
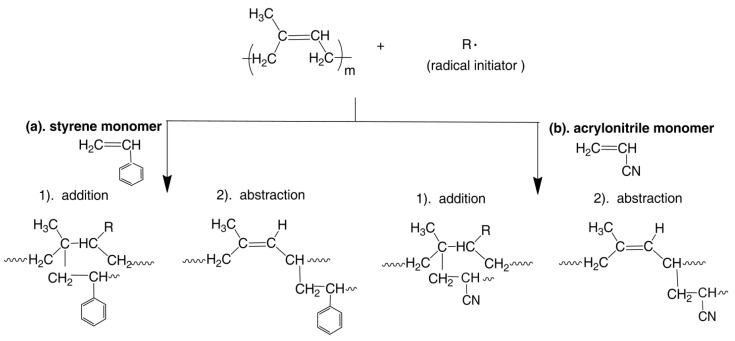
Possible structures of (**a**) DPNR-g-PS, (**b**) DPNR-g-PAN.

**Figure 2 polymers-13-03447-f002:**
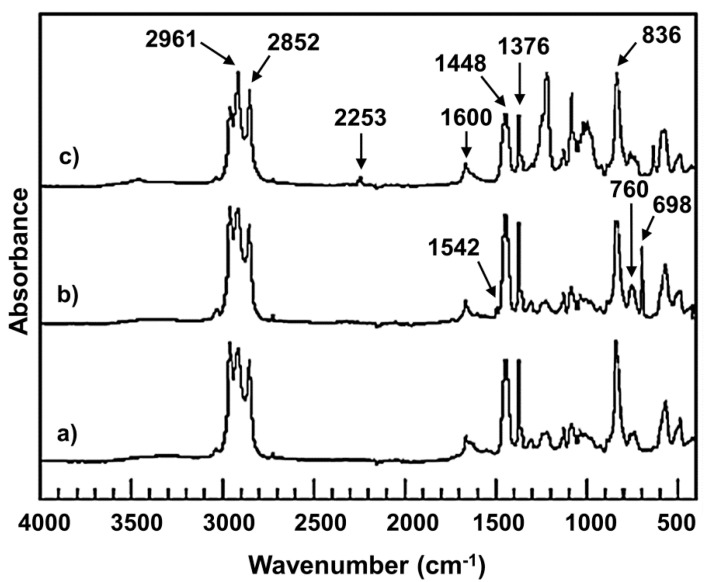
FTIR spectra of (**a**) DPNR; (**b**) DPNR-g-PS (15 phr); and (**c**) DPNR-g-PAN (15 phr).

**Figure 3 polymers-13-03447-f003:**
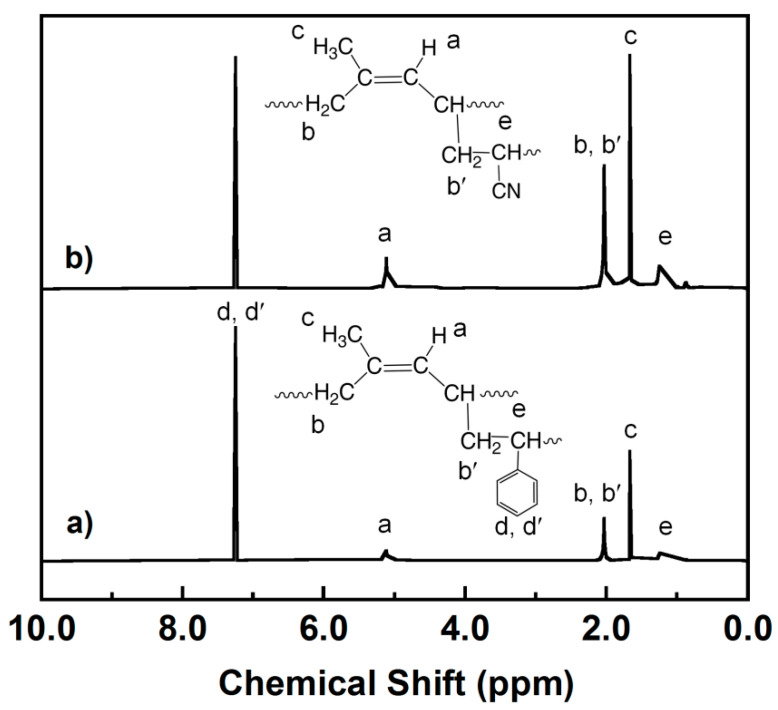
^1^H-NMR spectra of (**a**) DPNR-g-PS (15 phr) and (**b**) DPNR-g-PAN (15 phr).

**Figure 4 polymers-13-03447-f004:**
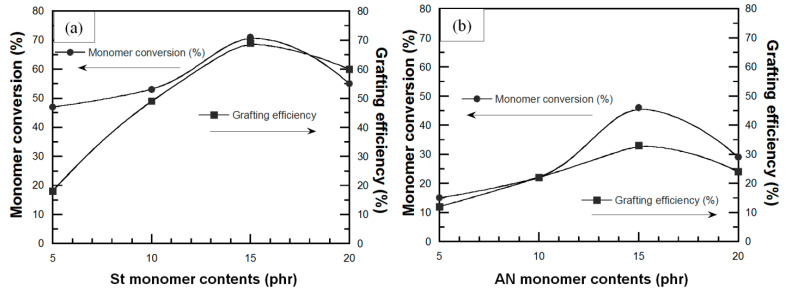
Variation of monomer conversion and grafting efficiency of (**a**) DPNR-g-PS as a function of St monomer content and (**b**) DPNR-g-PAN as a function of AN monomer content.

**Figure 5 polymers-13-03447-f005:**
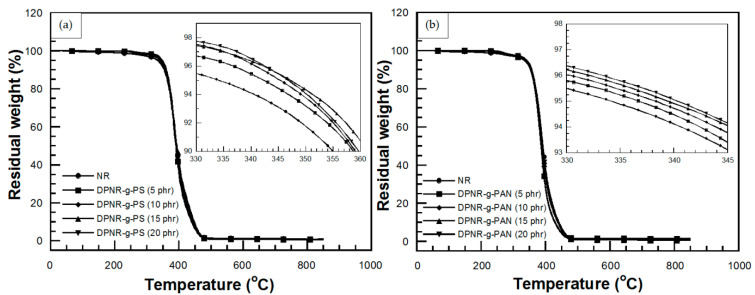
Degradation temperature at 5% weight loss of (**a**) NR and DPNR-g-PS at various St monomer contents and (**b**) NR and DPNR-g-PAN at various AN monomer contents.

**Figure 6 polymers-13-03447-f006:**
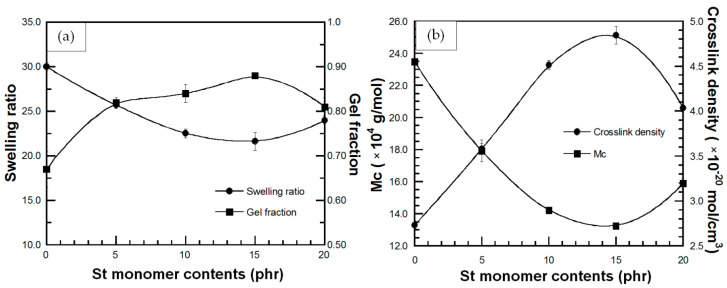
(**a**) (●) swelling ratio and (■) gel fraction; (**b**) (●) crosslink density and (■) molecular weight between crosslinks (M_c_) of DPNR-g-PS in toluene solvent at various St monomer contents.

**Figure 7 polymers-13-03447-f007:**
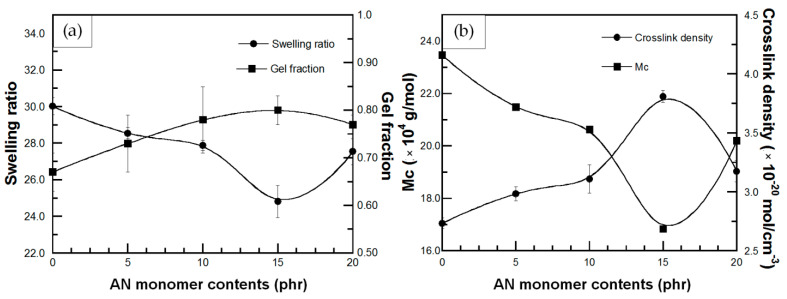
(**a**) (●) swelling ratio and (■) gel fraction; (**b**) (●) crosslink density and (■) molecular weight between crosslinks (M_c_) of DPNR-g-PAN in toluene solvent at various AN monomer contents.

**Figure 8 polymers-13-03447-f008:**
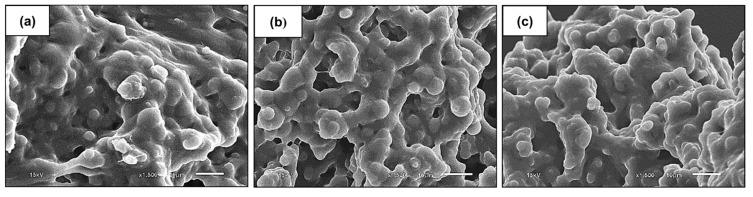
SEM micrographs (×1500 magnification) of powdered rubber of (**a**) NR; (**b**) DPNR-g-PS (15 phr), (**c**) DPNR-g-PAN (15 phr).

**Figure 9 polymers-13-03447-f009:**
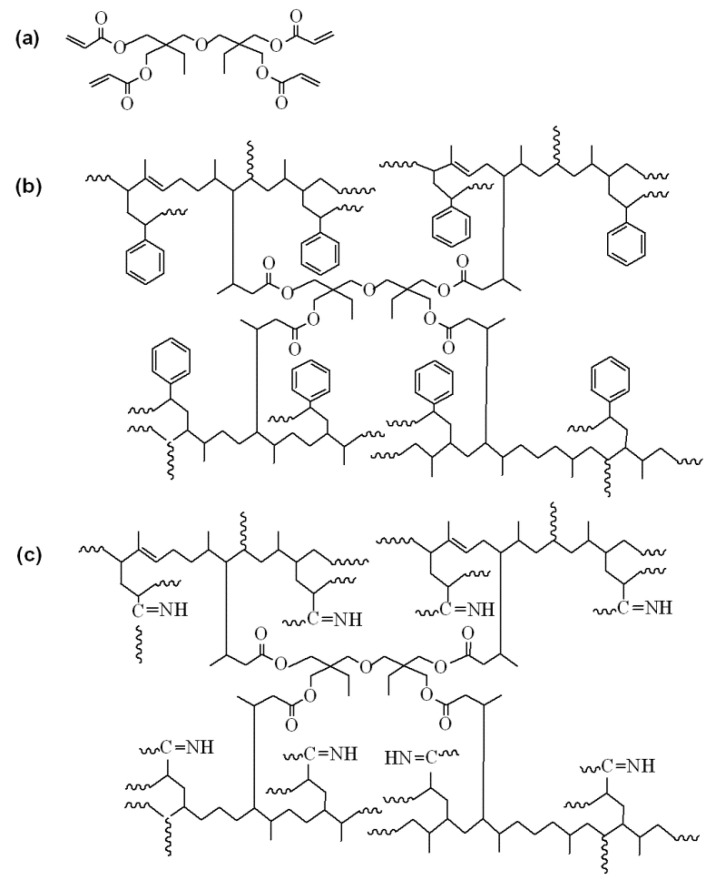
Chemical structure of (**a**) DTMPTA; (**b**) possible structure of crosslinked UFPNR-g-PS; (**c**) possible structure of crosslinked UFPNR-g-PAN.

**Figure 10 polymers-13-03447-f010:**
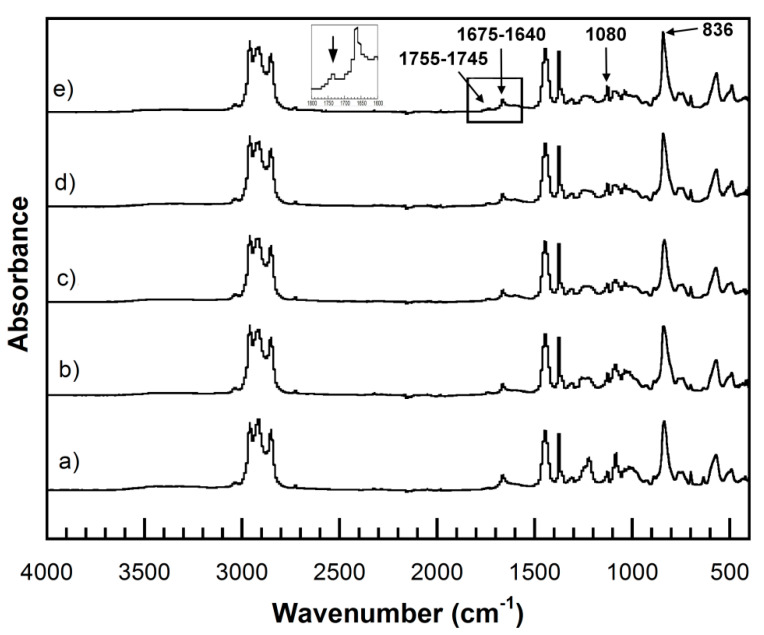
FTIR spectra of obtained UFPNR-g-PS (5 phr) at various radiation doses: (**a**) 0; (**b**) 50; (**c**) 100; (**d**) 200; (**e**) 300 kGy.

**Figure 11 polymers-13-03447-f011:**
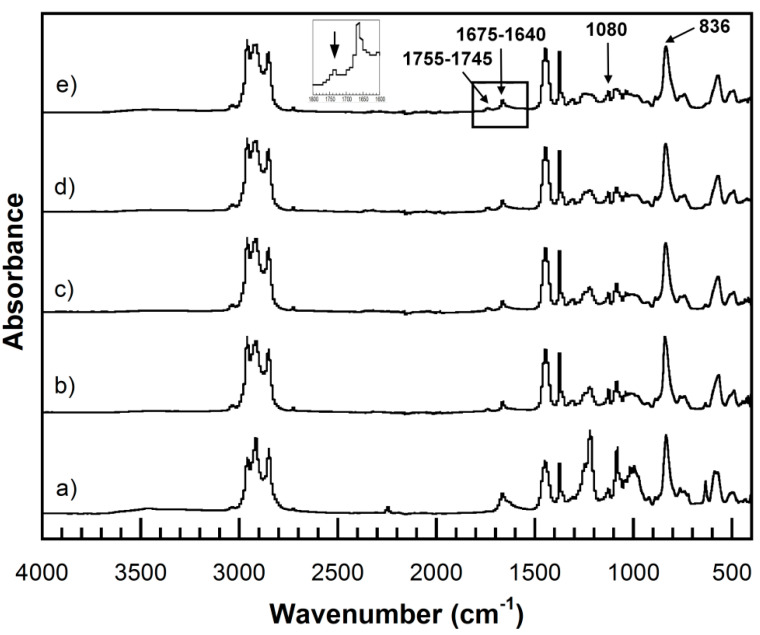
FTIR spectra of obtained UFPNR-g-PAN (5 phr) at various radiation doses: (**a**) 0; (**b**) 50; (**c**) 100; (**d**) 200; (**e**) 300 kGy.

**Figure 12 polymers-13-03447-f012:**
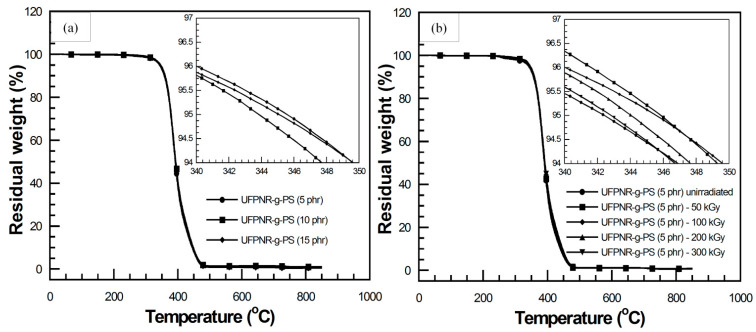
TGA thermograms of (**a**) UFPNR-g-PS at various St monomer contents irradiated at 100 kGy and (**b**) UFPNR-g-PS (5 phr) at various radiation doses.

**Figure 13 polymers-13-03447-f013:**
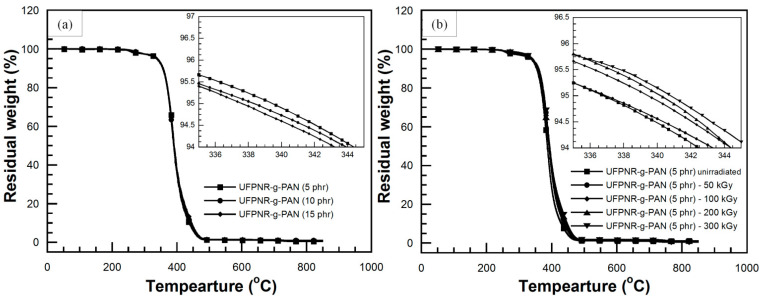
TGA thermograms of (**a**) UFPNR-g-PAN at various AN monomer contents irradiated at 100 kGy and (**b**) UFPNR-g-PAN (5 phr) at various radiation doses.

**Figure 14 polymers-13-03447-f014:**
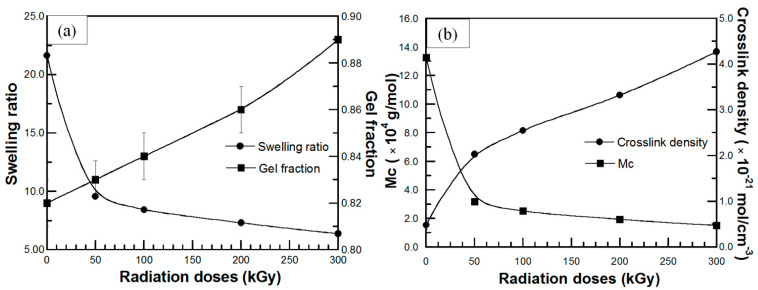
(**a**) (●) swelling ratio and (■) gel fraction; (**b**) (●) crosslink density and (■) molecular weight between crosslinking (M_c_) of UFPNR-g-PS (5 phr) in toluene solvent at various radiation doses.

**Figure 15 polymers-13-03447-f015:**
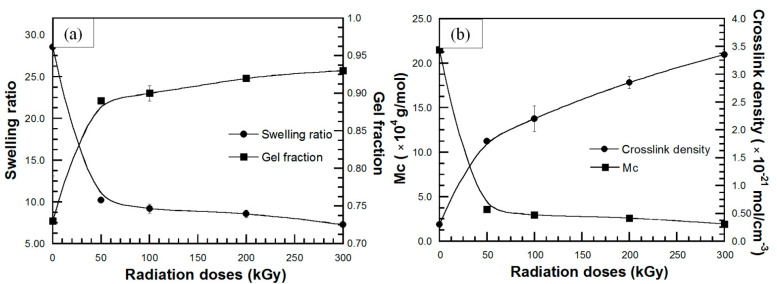
(**a**) (●) swelling ratio and (■) gel fraction; (**b**) (●) crosslink density and (■) molecular weight between crosslinking (M_c_) of UFPNR-g-PAN (5 phr) in toluene solvent at various radiation doses.

**Figure 16 polymers-13-03447-f016:**
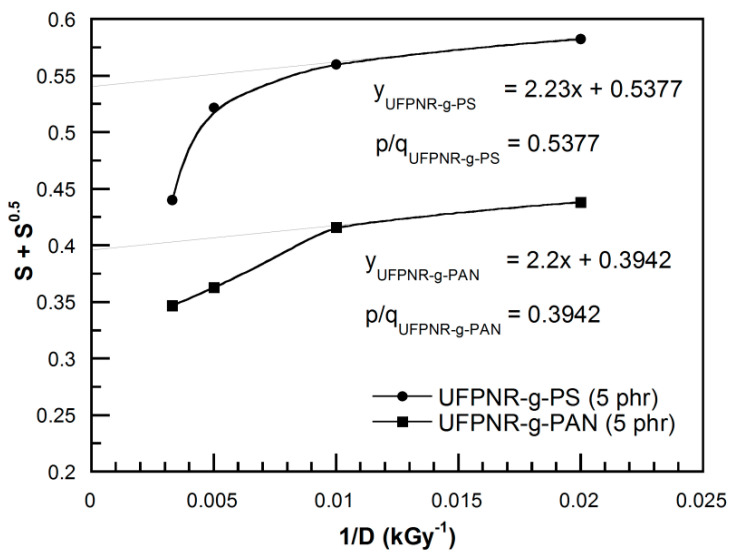
Sol-gel analysis of radiation crosslinking of UFPNR-g-PS (5 phr) and UFPNR-g-PAN (5 phr) at 50, 100, 200, and 300 kGy.

**Figure 17 polymers-13-03447-f017:**
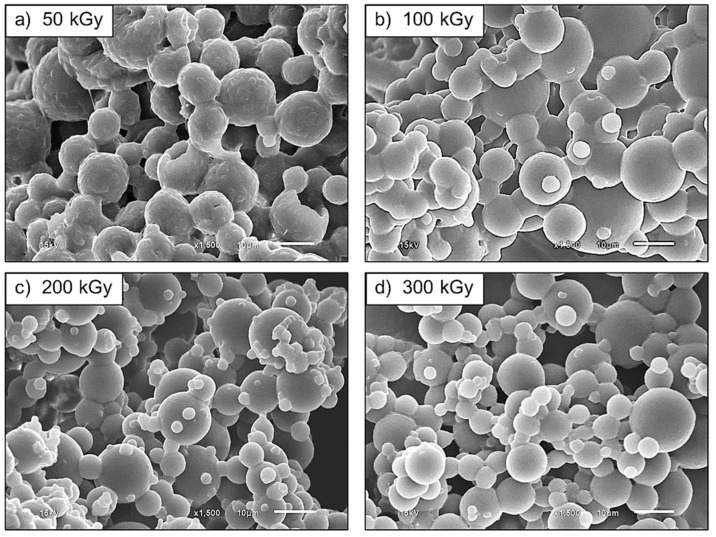
SEM micrographs (×1500 magnification) of UFPNR-g-PS (5 phr) at various radiation doses.

**Figure 18 polymers-13-03447-f018:**
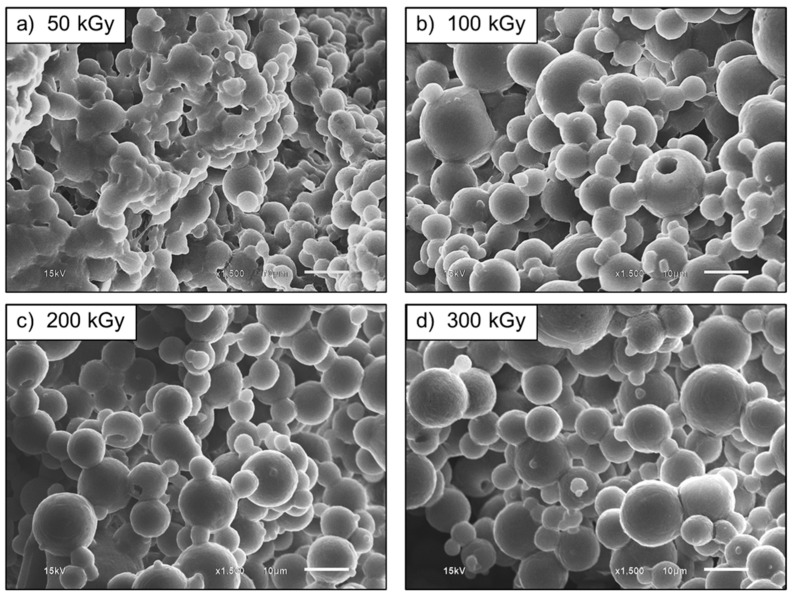
SEM micrographs (×1500 magnification) of UFPNR-g-PAN (5 phr) at various radiation doses.

## Data Availability

Data is contained within the article.

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
