# Peer review of "Radiation Graft-Copolymerization of Ultrafine Fully Vulcanized Powdered Natural Rubber: Effects of Styrene and Acrylonitrile Contents on Thermal Stability"

_polymers, 2021, doi:10.3390/polym13193447_

Round 1
Reviewer 1 Report
The paper presents a series of information on studies related to the preparation of UFPRs and the analysis of their physical, chemical and thermal properties. - the paper includes several results of interest to the scientific community:
- keywords must be rewritten and must not contain abbreviations; - at the end of the Introduction section the structure of the paper should be presented in more detail;
- - research methodology needs to be substantially improved; the objective of the research must also be presented. Thus, the decisions to take samples with a certain composition must be explained;
- - the macroscopic images must be presented together with the test pieces made;
- - in the case of Figures 8, 17 and 18 respectively, the resolution needs to be improved;
- - the discussion part should be improved so as to better highlight the novelty of the research compared to other research in the field;
- - the conclusions should be more concrete and include future research directions.
Author Response
Dear Reviewer
Please find the attached file, response to the referees
Best Regards,
Professor Sarawut Rimdusit, Ph.D.
Research Unit on Polymeric Materials for Medical Practice Devices,
and Polymer Engineering Laboratory
Department of Chemical Engineering
Faculty of Engineering
Chulalongkorn University
Bangkok 10330, THAILAND
Email: sarawut.r@chula.ac.th

Reviewer 2 Report
Ultrafine fully vulcanized powdered natural rubbers have been used as a toughening filler in a polymer matrix. Authors have made UFPRs by electron beam irradiation of the copolymers in the presence of ditrimethylolpropane tetraacrylate as crosslinking agent, followed by a spray drying process, and its chemical structure, thermal stability, and physical properties have been also investigated. However, the current form of this study cannot be acceptable. Some aspects as listed below:
1. What are the advantages of the application with NR-g-PS and NR-g-PAN?
2. In section 2, The title should be “Materials and Methods”?
3.In section 2.5, analysis methods and parameters of NMR should be given more details.
4. The image quality in Fig 2 should be improved.
5. In Fig 16, there is only one figure. (a) and (b) should be revised.
6. Why only the 5phr is studied for the effect of radiation dose on swelling behavior of UFPNR-g-PS and UFPNR-g-PAN? How about the 3phr or others? It can be further optimized the process and content parameters.
7. Why the average particle size of the UFPNR-g-PANs is similar? What is the relationship between the size and its performance?
Author Response
Dear Reviewer
Please find the attached file, response to the referees
Best Regards,
Professor Sarawut Rimdusit, Ph.D.
Research Unit on Polymeric Materials for Medical Practice Devices,
and Polymer Engineering Laboratory
Department of Chemical Engineering
Faculty of Engineering
Chulalongkorn University
Bangkok 10330, THAILAND

Reviewer 3 Report
Authors presented an extensive experimental procedure, but the discussion lacks of scientific soundness.
Page 1 – Line 30
The results revealed that the developed UFPNRs grafted styrene and acrylonitrile are promising candidates as toughening fillers in polymer composites.
Comment: UFPNRs were not tested as filler composites. No results were presented, which could justify this sentence. A proper highligth has to be presented.
Page 1 – Line 40
... nitrile–butadiene (UFPNBR) [5,6], etc.
Comment: “etc” cannot be used in substitution to references. Authors must exclude it and add references related to powdered rubbers.
Page 1 – Line 41
UFPRs were prepared by a combination of irradiation of rubber latex by gamma rays or electron beam in the presence of crosslinking agent, followed by a fast spray drying process.
Comment: Is it the only feasible method to produce UFPRs? Or was it (combination of EB and gamma radiation) used in this work? English must be fully revised by a native speaker.
Page 2 – Line 49
This is because the interactions between crosslinked rubber particles are much lower than the cohesion of bulk rubber particles, and provide fine domain with the same size and good dispersion, no matter how high is the blend ratio of UFPR to matrix.
Comment: Is it valid to any type of polymeric matrix?
Page 3 – Line 141
The latex mixture was stirred for 15 min before vulcanized by electron beam irradiation at the doses of 50, 100, 200, and 300 kGy (supported by Thailand Institute of Nuclear Technology (Public Organization)).
Comment: Authors mentioned in the introduction the use of gamma ray to produce UFPR. At this point, they describe the use of EB only. Why? Selected EB doses were based on what previous works?
Page 4 – Line 152
were characterized using FTIR spectroscopy (Perkin Elmer 2000 model) in the range of 400–4000 cm-1.
Comment: Resolution? Scans? ATR?
Page 5 – Line 226
The lower intensity peaks at 1600 and 836 cm-1 of NR-g-PS and NR-g-PAN compared with NR were attributed to consumption of double bonds during grafting process.
Comment: First, FTIR spectra were not normalized, it means, peak intensity of different spectra cannot be direct compared. Second, spectra resolution in Figure 2 do not allow to affirm the consumption of double bonds. Indeed, FTIR spectra no not confirm that PS and PAN were grafted onto NR.
Page 8 – Line 293
Grafting improves Td5 from 334oC for NR to 342, 345 and 347 oC for addition of PS content at 5, 10, and 15 phr. This is due to the interaction of PS and NR, along with high rigidity and stability of aromatic benzene ring in PS structure which has high degradation temperature in range of 375-450 oC
Comment: According to authors, the Td5 increasing of ~4% results from grafting, which is speculative. It could result from residual PS in rubber or, it can be experimental error. How many samples were tested fo each condition/sample? The same can be applied for PAN-containing samples.
Page 11 – Line 387
Comment: Discussion of FTIR spectra is speculative.FTIR spectra in Figures 10 and 11 do not have proper resolution (normalized?) to be applied on quantative discussion, or comparison of peak intensities.
Page 13 – Line 428
The effect of monomer content used on degradation temperature at 5% weight loss (Td5) of UFPNRs under 100 kGy was investigated. Fig. 12(a) showed Td5, i.e. 346 oC, 344 oC, and 345 oC of UFPNR-g-PS at ST content of 5, 10, and 15 phr, respectively. At higher ST contents, lower Td5 of UFPNR-g-PS was observed.
Comment: Changes of 1 or 2 °C cannot be taken as consequence of PS content, it does not make sense.
Page 13 – Line 455
Consequently, thermal stability of UFPNR-g-PS at 5 phr ST content was studied at various radiation doses as depicted in Fig. 12(b). The Td5 of UFPNR-g-PS (5 phr) at 0, 50, 100, 200 and 300 kGy was 342, 346, 346, 344, and 343 oC, respectively. An increase of the Td5 in the whole range of radiation doses was due to formation of a three-dimensional crosslinked network under the radiation influence.
Comment: Indeed, there is no meaningful Td5 change as consequence of EB dose increasing, as claimed by authors.
Author Response

(The authors gave the same response as above.)

Round 2
Reviewer 2 Report
Ultrafine fully vulcanized powdered natural rubbers have been used as a toughening filler in a polymer matrix. Authors have revised the manuscript according to the commands. The current form of this study can be acceptable after minor revision. Some aspects as listed below:
- In page 4 line 178-182, variables in the equation should be italicized.
- In Line 521-828, ……
There are still some errors of grammar, and punctuation in the manuscript. It is recommended that a native English speaker correct this document throughout.
Author Response
Dear Reviewer,
Please find the authors’ responses to reviewers in the attached file
Best Regards,
Professor Sarawut Rimdusit, Ph.D.
Research Unit on Polymeric Materials for Medical Practice Devices,
and Polymer Engineering Laboratory
Department of Chemical Engineering
Faculty of Engineering
Chulalongkorn University
Bangkok 10330, THAILAND
Email: sarawut.r@chula.ac.th

Reviewer 3 Report
Accept in current form.
Author Response

(The authors gave the same response as above.)
